

# Field Test of Wake Steering at an Offshore Wind Farm

Paul Fleming[1], Jennifer Annoni[1], Jigar J. Shah[2], Linpeng Wang[3], Shreyas Ananthan[2], Zhijun Zhang[3], Kyle Hutchings[2], Peng Wang[3], Weiguo Chen[3], and Lin Chen[3]

[1]National Wind Technology Center, National Renewable Energy Laboratory, Golden, CO, 80401, USA
[2]Research & Development, Envision Energy USA Ltd, Houston, TX, 77002, USA
[3]Research & Development, Envision Energy Limited, Shanghai, 200051, China

*Correspondence to:* Paul Fleming (paul.fleming@nrel.gov)

**Abstract.**

In this paper, a field test of wake steering control is presented. The field test is the result of a collaboration between the National Renewable Energy Laboratory (NREL) and Envision Energy, a smart energy management company and turbine manufacturer. In the campaign, an array of turbines within an operating commercial offshore wind farm in China have the normal yaw controller modified to implement wake steering according to a yaw control strategy. The strategy was designed using NREL wind farm models, including a computational fluid dynamics model, SOWFA, for understanding wake dynamics and an engineering model, FLORIS, for yaw control optimization. Results indicate that, within the certainty afforded by the data, the wake-steering controller was successful in increasing power capture, by amounts similar to those predicted from the models.

## 1 Introduction

Wind farm control is an active field of research in which the controls of individual turbines co-located within a wind farm are coordinated to improve the overall performance of the farm. One objective of wind farm control is improving the power production of wind farms by accounting for the wake interactions between nearby turbines.

In one wind farm control concept, turbines are yawed to introduce a deflection of the wake away from downstream turbines. This method has been referred to as "controlling the wind" Wagenaar et al. (2012) and "yaw-based wake steering" Fleming et al. (2014b). High-fidelity simulations of wake steering have shown the potential of this technique. Jiménez et al. (2010) used computational fluid dynamics (CFD) simulations to demonstrate the wake deflection capability of wind turbines and provided a model of this deflection. In Fleming et al. (2014b), they used NREL's CFD-based Simulator fOr Wind Farm Applications (SOWFA) to investigate the capabilities of wind turbines to redirect wakes. In Vollmer et al. (2016), the behavior of wake steering in different atmospheric conditions was investigated also using CFD. Finally, in Fleming et al. (2014a), simulations of two-turbine wind farms, again using SOWFA, were used to show that through wake steering, the net power of the two turbines is increased when the upstream turbine applies an intentional yaw misalignment.

Based on high-fidelity simulations, there appears to be good opportunities for improved power performance of wind farms with significant wake losses. Recent efforts have focused on the design of lower-fidelity, controller-oriented models, and con-



trollers based on these models that use wake steering to actively improve power. In Gebraad et al. (2014), the FLOw Redirection and Induction in Steady State (FLORIS) model is described and used to determine optimal yaw settings for a model six-turbine wind farm. Set points for a particular wind speed and direction are determined by optimizing the yaw angles of the turbines using FLORIS, and these set points are used in SOWFA simulations. The results from SOWFA agree with the predictions from

FLORIS, and total power capture is increased 13%. This work is carried further, and FLORIS is used to assess the overall improvement from control, first for one speed and over a wind rose of directions in Fleming et al. (2015) and then to determine the overall annual energy production in Gebraad et al. (2016). These studies indicate a good potential for improved overall annual power production for wind farms experiencing significant wake losses. It should be noted that even greater benefits can be yielded if future wind farms are designed for active control of wakes, rather than using large inter-turbine spacings to avoid

wake losses. This combined optimization of wind farm control and wind farm system engineering is a subject of active research. These simulation studies have demonstrated a theoretical potential of wind farm control. However, it is often noted that issues arising in implementation in real conditions might undermine the positive results. Inaccuracies in the control-oriented models or high-fidelity simulations have been cited as a potential issue. Additional modeling of constantly changing wind direction could improve the comparison between simulation and field testing.

Some field testing of wake steering has been performed to date to understand the potential of this wind farm control strategy. In Wagenaar et al. (2012), wake steering is implemented at a scaled wind farm; however, the results are inconclusive. Wind tunnel testing of wake steering is performed in Schottler et al. (2016) and Campagnolo et al. (2016), and the results are encouraging because in each case wake steering is observed, overall power capture is improved for two-turbine cases. The results are in alignment with earlier simulation studies conducted by Jiménez et al. (2010),Gebraad et al. (2014),and Fleming

et al. (2014a) in the amount of improvement and the asymmetric relationship of yaw misalignment and power improvement. At the full scale, a nacelle-mounted lidar is used to observe wake deflection on a utility-scale turbine in Trujillo et al. (2016). Finally, in Fleming et al. (2016), two ongoing studies of wake steering are presented, which are both part of a multiyear U.S. Department of Energy Atmosphere to Electrons project. In one of the field studies at the Scaled Wind Farm Test Facility (SWiFT), two 27-m-diameter [D] turbines (intentionally aligned in the dominant wind direction at a spacing of 5 diameters)

were used to perform a comprehensive test of wake steering. A second ongoing experiment was done in which a rear-mounted lidar was used to monitor a wake of a utility-scale turbine, which is set to hold a specific yaw misalignment for a prolonged period of time. Initial results from both campaigns are in accord with the model predictions described earlier.

In the present study, a wind farm controller that performs wake steering is designed and implemented for an operating commercial offshore wind farm in China with Envision turbines. The control strategy is simple and based on the approach used

to control the simulated wind farm in Gebraad et al. (2014). The test was run for several months and data were collected and compared from time periods when the controller was operating and not operating. The results and analysis indicate a successful improvement in power production. Additionally, the data provide important validation of the models, specifically SOWFA and FLORIS, used in the design of the controller, and can be generalized to similar CFD and control-oriented models. There are some qualifications to these results that will be fully considered in the text.



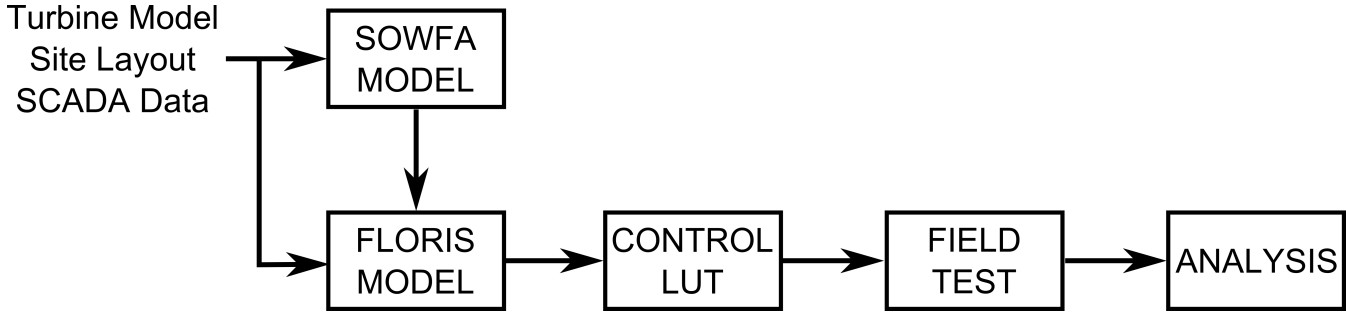

**Figure 1.** This figure demonstrates the workflow of this project. In particular, this project started by running CFD models of the turbines in this field study to understand the wake characteristics. The FLORIS model was tuned to those CFD simulations. A control look-up table was generated that contained the optimal yaw settings for the control turbine. A field test was conducted with the controller on and off. The data was analyzed and the results are presented in this paper. (LUT) - look-up table and (SCADA) - supervisory control and data acquisition

The contributions of this paper include the results and analysis from a wind farm test of wake-steering control. The project is a collaborative project between the National Renewable Energy Laboratory (NREL) and Envision Energy, a smart energy management company and turbine manufacturer. The positive results motivate further encourage into the design and development of such control. Additionally, the paper provides feedback on the performance of wind farm control modeling tools in their ability to predict the effect of wind farm control strategies.

## 2 Project Overview

The goal of the project was to implement a test of yaw-based wake steering at an operating wind farm. The project was broken down into a workflow illustrated in Fig. 1, which shows the stages of work and the structure by which this paper is organized.

The first stage of work is the selection of a wind farm for use in the experiment. The Longyuan Rudong Chaojiandai offshore wind farm in Jiangsu, China, consists of turbines from multiple manufacturers undergoing various phases of construction. A selected portion of the site was studied for this effort, consisting of 25 Envision EN136/4 MW turbines incorporating a high-speed three-stage gearbox and induction generator (see Fig. 2 (left)).

From the wind farm, a subset of turbines was selected for implementing the experiment. These turbines form the front two rows of turbines for winds coming from the northeast. The arrangement of these turbines and their names to be used throughout the paper are shown in Fig. 2.

For the campaign, a single turbine was selected to be the controlled turbine. This is turbine C1, indicated in Fig. 2. The turbine is shown to wake three particular turbines: D1, from a wind direction of $340^o$ at a distance of 7D, D2, from a wind direction of $51^o$ at a distance of 8.6D, and D3, from a wind direction of $81^o$ at a distance of 14.3D. A control strategy was designed to improve the summed power of turbine C1 and a downstream turbine (i.e., C1 and D1, C1 and D2, and C1 and D3), through yaw misalignment of turbine C1. This approach provides three tests of wake steering at different interturbine distances. Of these, the pairing with turbine D1, at 7D, is most promising. This is the distance used in Fleming et al. (2014a),





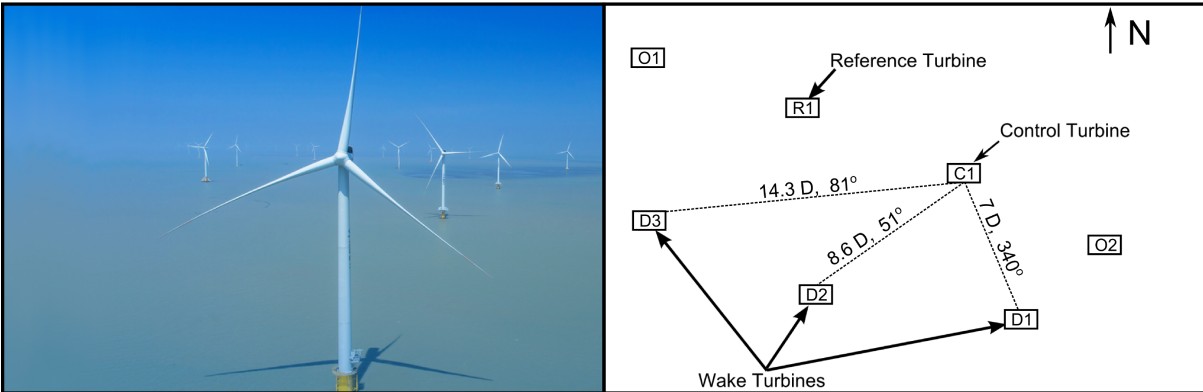

**Figure 2.** (Left) Rudong Wind Farm used in the field study. (Right) Turbine locations. The wake-steering control strategy is implemented to mitigate the wake interactions between C1 and (D1, D2, and D3). As indicated by the diagram, C1 is the control turbine, R1 is the reference turbine, and D1, D2, and D3 are the turbines that are down-stream turbines waked by C1. Turbines O1 and O2 are other turbines not directly used in the study but whose wakes are noticed in certain directions.

which showed a potential net improvement of 4.5% in power capture in a study with the NREL 5-MW reference turbine for the waking wind direction. Finally, a reference turbine which is not waked in any of the experimental directions is chosen to provide reference signals. This is turbine R1 in Fig. 2.

In the initial phase of work, Envision shared with NREL a FAST model of the turbines (FAST is an aero-servo-elastic wind
turbine simulation tool maintained by NREL, Jonkman and Buhl Jr. (2005)). Additionally, Envision provided the layout of the turbines used in the campaign. Finally, Envision provided several months of supervisory control and data acquisition (SCADA) data used to evaluate and tune models in advance of the campaign. As shown in Fig. 1, these inputs were then included in the derivation of the models used in the yaw control strategy, which will be described in the next section.

## 3   Modeling

### 3.1   SOWFA

The first model to be used in this study was the SOWFA model, Churchfield and Lee (2014). SOWFA is a wind farm simulation tool, which models the atmospheric boundary layer using CFD, and then models the turbines using embedded FAST models (in the version of SOWFA used in this work). The turbines and flow interact through a two-way actuator line coupling.

SOWFA has been validated against wind farm SCADA data, Churchfield et al. (2012). However, validating SOWFA is not
the focus of this campaign. The primary use of SOWFA in this study is generating data sets of wakes simulated from the Envision turbine, which can be used to tune the FLORIS model (and repeating the procedure that was done for the NREL 5-MW reference turbine in Gebraad et al. (2014).





Using the FAST model of the turbine, a suite of simulations is assembled. A wind condition of 8 m/s wind and 6% turbulence, Gebraad et al. (2014), was used for this study. Simulations were then run with a single turbine operating with various amounts of yaw misalignment, as well as for two turbine cases, wherein the upstream turbine has various yaw misalignments and the downstream turbine is placed in various positions downstream and cross stream. The trends in power for the single-turbine

case and two-turbine case provide important data with which to tune the control-oriented FLORIS model, described in the next section.

## 3.2 FLORIS

Using a FAST model of the turbine and the data sets produced by SOWFA, it is possible to obtain a FLORIS model that can predict the power of turbines in a farm in steady state including wake redirection. As discussed in Gebraad et al. (2014), the

FLORIS model is primarily based on the Jensen model, Jensen (1984), and the Jiménez model, Jiménez et al. (2010). In particular, FLORIS identifies three different wake zones with separate wake recovery parameters to capture the wake characteristics. In addition, FLORIS includes the Jiménez model to incorporate the wake deflection caused by yaw misalignment. FLORIS contains a set of parameters to be tuned for a given turbine, including parameters describing the turbine and wake behavior.

Some parameters of the FLORIS model can be set directly using the FAST model. Examples of this include the rotor radius

and the table of power and thrust coefficients by wind speed, which can be obtained by running steady wind simulations in FAST.

The remaining parameters are tuned in this work through an optimization routine that minimizes the error between the power outputs simulated in SOWFA and those predicted by FLORIS. As a design rule, the smallest set of parameters should be adjusted away from their default settings that produce a reasonable fit. Through iteration, it was determined that adjustments

to four parameters gave a good approximation. The parameters that we focus on are described below.

The parameter $pP$ relates yaw misalignment to reduction in power by:

$$yawLoss = cos(\gamma)^{pP} \tag{1}$$

where $\gamma$ is the yaw misalignment of the turbine. The higher $pP$ is, the more quickly a turbine loses power by misalignment, and the less likely wake steering will work, because the downstream turbine needs to recover more power to compensate for

power lost by the upstream turbine.

The parameter $k_e$ determines the rate at which a wake expands and recovers to the free stream velocity. A larger $k_e$ value indicates a faster wake recovery to free stream. Standard values in literature range from 0.05 to 0.1. Similarly, $k_d$, based on the Jiménez model, describes the rate at which a deflected wake reverts to the free-stream direction. A larger $k_d$ parameter indicates that the wake is less sensitive to yaw misalignment. Standard values in literature range from $k_d = 0.1$ to $k_d = 0.3$.

Finally, $initWD$ describes an initial wake deflection angle without steering, which is important for capturing the asymmetry of wake steering. This asymmetry is likely caused by the combination of the rotation of the turbine and the shear layer in the atmospheric boundary layer.

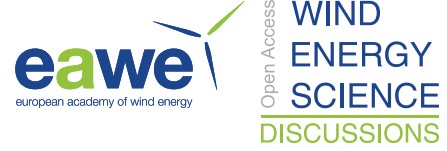

**Table 1.** FLORIS Wake Tuning

| Parameter | NREL 5MW | Envision |
|-----------|----------|----------|
| $K_d$ | 0.17 | 0.26 |
| $K_e$ | 0.05 | 0.063 |
| $initWD$ | -4.5 | -3.7 |
| $pP$ | 1.88 | 1.43 |

The results of the tuning optimization are presented in Table 1, which shows both the default values obtained from tuning to the NREL 5-MW reference turbine, as well as the newly obtained values.

Among these, the lower value of pP is interesting, as 1.43 is below the value obtained for the NREL 5-MW turbine (1.88), and other experimental results. For example, the coefficient is fit to wind tunnel tests in Medici (2005) to be 2.0. However, it is an attractive number as it implies that wake steering can be performed with less losses incurred on the upstream yawed turbine.

## 4    Control Design and Field Test

Given a completed FLORIS model, it is now possible to derive a set of yaw misalignments for turbine C1 that will optimize power for the pairs of turbines (D1, D2, and D3 downstream) by wind direction. Wind speed is not used as an input as inspection indicated minimal sensitivity. It is important to note that there is not much benefit at very low and very high wind speeds, suggesting that it is sufficient to enable and disable the controller by wind speed rather than scheduling.

Some constraints were placed on the optimization. First, for turbine loading and safety reasons, the maximum yaw misalignment was limited to 25 degrees. Second, it was decided for this experiment to limit the controller to positive yaw misalignment angles (in our nomenclature this is rotating the turbine counter-clockwise of the wind). This is because it has been demonstrated to be more effective, and including negative misalignments might raise loads and require a nontrivial transition from positive to negative yaw misalignments near the wake cross-over point, i.e., when it becomes more beneficial to redirect the wake from the right side of the downstream turbine to the left side of the downstream turbine. Using the tuned FLORIS model and these constraints, it is possible to now derive an optimal table of yaw misalignments for turbine C1. This is shown in Fig. 3.

The optimal yaw misalignment angles for turbine C1 are shown in the upper left of Fig. 3. The dashed lines indicate the directions in which turbine C1 wakes one of the three downstream turbines (see Fig. 2). Near the fully waked directions, the misalignments are largest and taper down as less deflection is needed to remove the partial wake overlap situations. The power loss of turbine C1, shown in normalized power, is indicated in the middle left plot. The overall "plant" gain for these four turbines is then shown in the lower left. Finally, the right figures show normalized power of the three downstream turbines with and without wake steering. Based on the percent improvement, the sum power is expected to increase for all three pairings, meaning the gains downstream exceed the losses upstream. This is least so for turbine D3, however, as at 14.3D, the baseline wake loss is much less, which is expected.

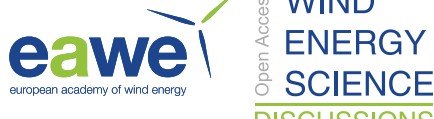



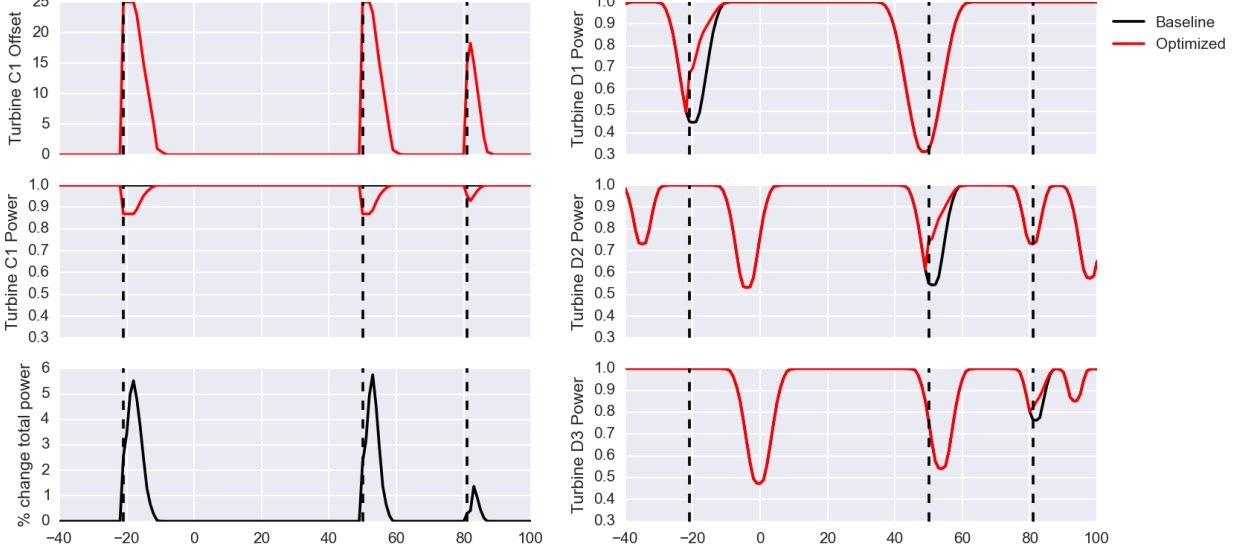

**Figure 3.** FLORIS optimal yaw misalignment results. The dotted vertical line indicates the direction in which the downstream turbine is fully waked by turbine C1. See Fig. 2 for details on the precise wind directions. Specifically, the first dotted line (from left to right) refers to the direction of turbine C1 fully waking downstream turbine D1. The second dotted line refers to the direction of turbine C1 fully waking downstream turbine D2. Finally, the third dotted line refers to the direction of turbine C1 fully waking downstream turbine D3. Wakes from other turbines than C1 are also evident off the dashed lines.

Using this table of offsets by wind direction, engineers at Envision modified the yaw controller of turbine C1 to deliver these offsets. Note, this offset tracking must be done within the limits of yaw control actuation, and for safety reasons the offset was disabled in sustained winds above 10 m/s. We note here also that the purpose of the experiment was to demonstrate the principle of wake steering and not produce a fully optimized closed-loop control implementation, which is expected to be the subject of future work. As will be discussed, the present controller offsets correctly in the average sense, with a wide variation of offsets occurring dynamically.

Following implementation of the controller, the experimental campaign was then run in two phases. In the first phase, the wind farm was operated normally while data relevant for this campaign was collected. This phase lasted 4 months from April 3, 2016, to August 5, 2016. The second phase used the controller designed above and was run an additional 4 months from August 5, 2016, to December 2, 2016.

It is regrettable that the campaign could not be run longer. Naturally, only a portion of the data features the wind directions of primary interest. Additionally, it should be noted that it would be better to alternate the controller on and off regularly throughout the campaign to better compare conditions when the controller is on than when it is off. However, Rudong is a



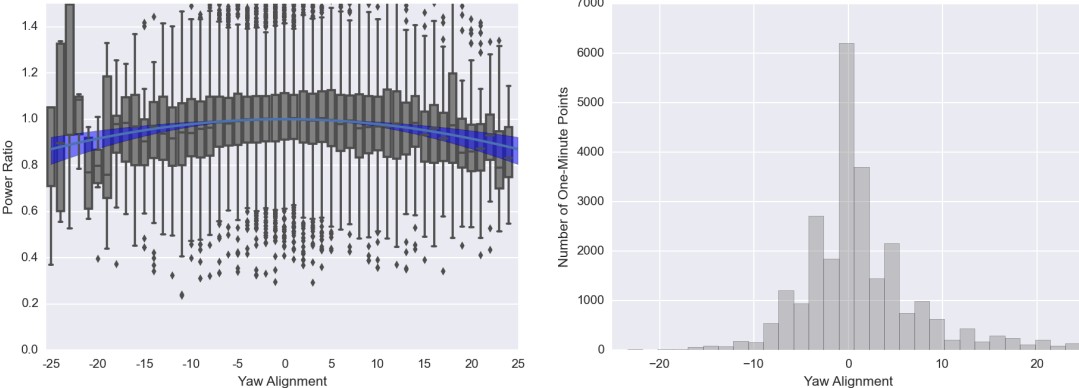

**Figure 4.** Cosine exponent fit. The left figure shows the range of data over various yaw misalignments and the power ratio. The blue line indicates the average of the data and the $pP$ value is determined from this average, $pP = 1.41$. The right plot indicates the number of points, included in each yaw alignment position.

commercial wind site with internal and external restrictions, and it was necessary to run the test within these constraints. It is important to note that there are limitations with field testing and data collection. In particular, the results are impacted because of the relatively short window of data collection. Further, the sequential testing pattern opens the possibility of confounding influences such as seasonal variation in atmospheric conditions. The results of these limitations will be discussed more in the
analysis section.

## 5   Results and Analysis

In this section, the results of the campaign are presented and analyzed. As a first step, the behavior of the upstream turbine can be analyzed independently. Specifically, the $pP$ term, which describes the loss of power against yaw misalignment, can be derived from the experimental data and compared with the value computed from SOWFA and used in FLORIS.

The data is collected at a 20-Hz rate from the campaign and is reduced to 1-minute averages. Next, we define a power ratio, which is the power of turbine C1 (the controlled turbine) divided by the power of turbine R1 (the reference turbine). We then consider this ratio as a function of the yaw misalignment of turbine C1. This misalignment is measured by the wind vane of turbine C1. Note that this calculation was repeated, wherein the misalignment is the difference between the yaw angle of turbine R1 and C1, without substantially affecting the results. The value of $pP$ is determined as the result of a minimization

problem between the data and Eq. (1). Additionally, to provide a form of confidence interval, a separate $pP$ is similarly derived for every day of data, and the $25^{th}$ and $75^{th}$ percentile values are used as a range of confidence. The data, fit, confidence interval, and amount of available data, are shown in Fig. 4.





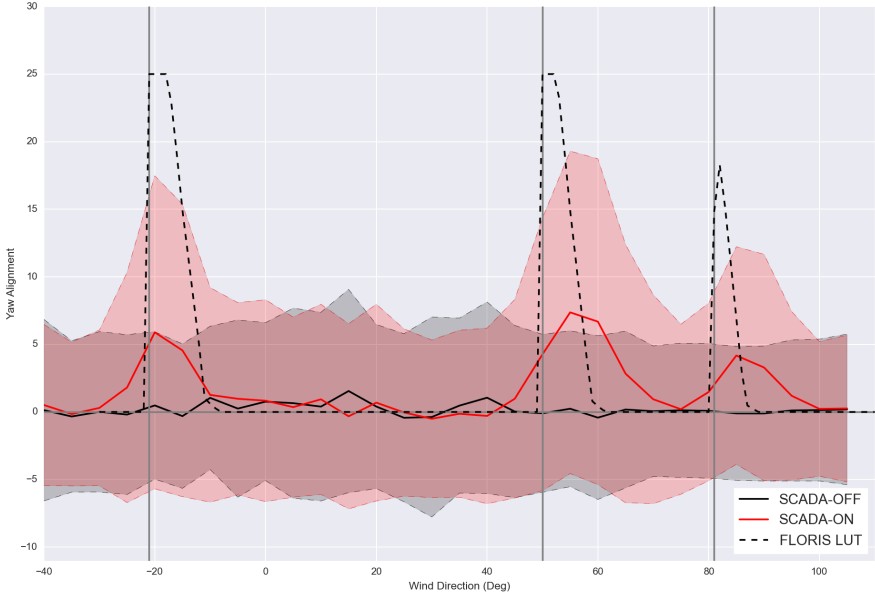

**Figure 5.** Statistical analysis of the SCADA vane data from turbine C1, with the offset controller off and on. This graph shows the controller of turbine C1 operating to redirect the wake away from the downstream turbines. The black and red lines indicate the average yaw offset positions from SCADA data for the baseline and offset strategy, while the dashed lines are those of the Look-Up Table (LUT) from FLORIS. Finally, the red and black regions show the range of points from the $25^{th}$ through $75^{th}$ percentiles for the baseline and offset SCADA data.

Using this method, the determined value of $pP$ was 1.41, which is close to the value derived from SOWFA. This is the first encouraging result, as the value from SOWFA initially appeared lower than expected. From a wake-steering perspective, it is useful that the upstream turbine will lose less power when operating in yawed conditions with a smaller exponent.

Fig. 5 shows the performance of the controller in obtaining the desired offsets. In the figure, for the baseline and optimized control strategies, the mean offset of turbine C1 is shown, as well as a shaded region including points (1-minute averages) from the $25^{th}$ through $75^{th}$ percentile of observed points. Finally, the initially prescribed pattern of FLORIS is shown. What can be observed is that implementing the offset strategy in real conditions where the turbine must track an ever-changing wind direction within the bounds of the yaw controller results in the sharp narrow peak of the optimal strategy being smoothed and spread. This implies that the actual control is somewhere between the baseline and optimal in terms of performance.



Finally, Figs. 6, 7, and 8 compare the overall performance across the directions for the three two-turbine pairings. To compute this performance, the following procedure is used. First, for each wind direction (in 5-degree bins), and for each turbine, for each day, a power curve of the form

$$P = min(aNv^3, P_{rated}) \tag{2}$$

is computed using an error-minimization technique, where $a$ is the fitted value and $N$ is the nominal value, which includes the air density, coefficient of power, rotor area, and efficiency losses. Therefore, $a$ represents a scalar gain on the power curve below rated, and also effectively shifts the rated wind speed. $v$ is the wind speed measured by the reference turbine R1.

This method, in which the data is used to derive a reduction value to the power curve was selected after significant effort to combine the available data into a complete analysis. Note that alternative methods, such as directly comparing power production

of the turbines was performed and gives similar results. The power curves are fit on a per-day basis (and not for example globally or through random boot-strapping) to help visualize the day-to-day variation in performance.

The computed value $a$ is the amount by which the power curve is reduced below nominal as observed for nonwaked and nonyawed turbines. From these values computed for each available day, for each control setting, the $25^{th}$, $50^{th}$, and $75^{th}$ percentile values are determined and used as the middle fit and confidence range. The reason for not computing the power

curve globally was to limit the impact of particular outlier days, and give some indication of the range of results observed. Where the range is small and the number of days of data collected large, the trends are converging. Finally, the predictions of FLORIS are overlaid on the plots for comparison. The values from FLORIS were computed using the same fitting approach to include some of the effects such as blurring between wind directions in the 5-degree bins.

For a first analysis, consider the turbine C1-D1 pair in Fig. 6. In particular, the results shown in Fig. 6 show the results

from FLORIS with no wake steering (FLORIS-BASE, shown with the dotted black line), FLORIS with optimal yaw set points (FLORIS-OPT, shown with the dotted red line), the field test with no wake steering (SCADA-BASE, shown with the solid black line), and the field test with the optimal yaw set points (SCADA-OPT, shown with the solid red line). For the upstream turbine C1, it can be noted that no trend in power change can be strongly observed. This is fairly consistent with the limited power loss of the lower $pP$ exponent, and the fact the achieved yaw misalignments were, on average, not very large. The results

for the 7D downstream turbine D1 are encouraging. A clear increase in power production is observed in the wind directions near the primary wake direction of -20 degrees (or 340 degrees). In the case of the main wake direction, the power is increased from 0.59 to 0.76 (an increase of 29%), which is less than the gain predicted by FLORIS of approximately 40%. However, it is evident that the smearing that results from various wind directions, which was evident in Fig. 5, is impacting the results as well. The overall pattern is again of a less dramatic gain; however, it is spread over a wider area. This result fits with the

analysis reported in Gaumond et al. (2013), which observed that the Jensen model (and by extension Jensen-based models like FLORIS) can predict wind farm power production more accurately if wind direction measurements are assumed to be uncertain. Incorporating this uncertainty into the FLORIS model should yield a better fit in that the wake deficits would appear





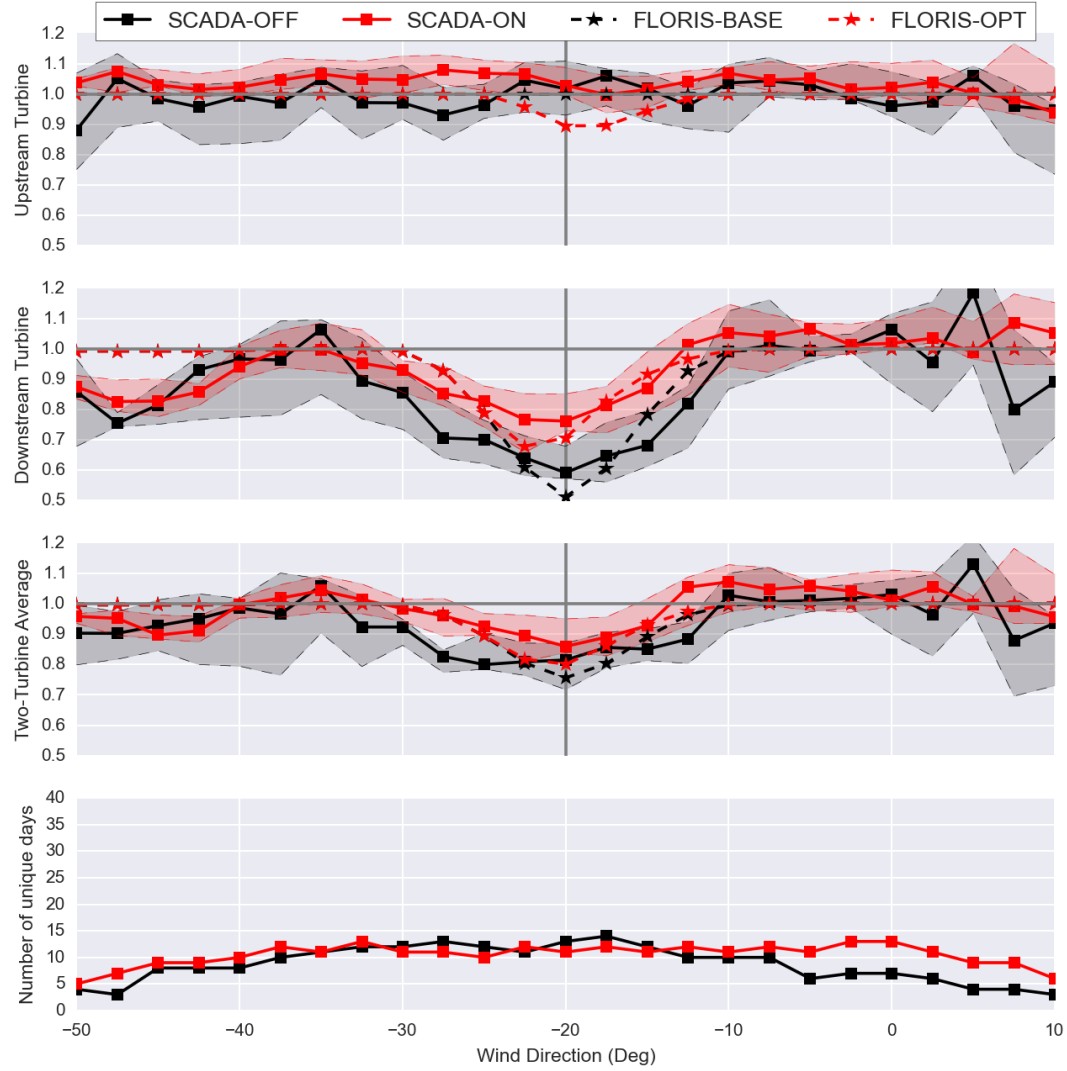

**Figure 6.** This figure shows the results fitting power curves for the turbine C1 to turbine D1 pair. FLORIS-BASE refers to the results from the FLORIS model with no wake steering and FLORIS-OPT refers to the results from the FLORIS model with the optimal yaw set points. SCADA-BASE refers to the field test where the controller was off and SCADA-OPT refers to the field test where the controller was on. The banded region shows range of fit values from the $25^{th}$ to $75^{th}$ percentile of fitted power curves. In particular, the top plot shows turbine C1 and the resulting power with the controller on (red) and off (black). The second plot shows the downstream turbine, i.e., turbine D1, with the offset controller of C1 on (red) and off (black). The third plot shows the overall power between the pair of turbines. There is a significant power gain when the controller is on versus when it is off. Finally, the bottom plot indicates the amount of days used to compute the statistics provided in the top three plots.



more widely spread as they do in the SCADA data. Incorporating this uncertainty into control design is the subject of ongoing research.

Turning to the power average of the two turbines, the lack of loss upstream and a good improvement downstream leads to a better-than-expected return for the pair. In interpreting Fig. 6, it is perhaps useful to focus as much on the shaded regions
(which indicate the region containing the $25^{th}$ to $75^{th}$ percentile of fitted power curves), as much as the square-lines, which are the $50^{th}$ percentile. Where the banded regions overlap least, is where we see a persistent change in performance. Large changes in the line, for example around 5 degrees in the downstream plot (Downstream Turbine) Fig. 6, are not yet meaningful as the regions are completely overlapping. That the banded regions of the two-turbine average contain significant non-overlapping regions around the wake control direction is probably a main positive finding of the paper. It suggests the power improvement
is consistent.

Finally, the bottom plot in Fig. 6 shows the number of separate days available for computing the data in each bin. It is useful to note that these do not indicate full days, but that some data were collected on a given day, which was in this direction and a separate power curve computed. A value of 10, for example, indicates that 10 days were used for a given bin for a given controller (sharing no data points), and these values were used to produce the statistics shown in the plot.
In the wind directions in which wake control is not active (to the left and right of the plot), there is no persistent trend between the baseline and optimized control on the downstream turbine. This is also mostly true where the turbine is waked by the reference turbine R1 at -47 degrees. This helps to confirm that the change observed is caused by wake deflection, rather than the atmosphere being different between testing periods being the cause of the underlying changes in wake behavior. This observation could be made for turbine D2 and 80 degrees in Fig 7, when it waked by turbine O2 (refer to Fig. 2), and turbine
D3 at 55 degrees in Fig. 8, when it is waked by reference turbine R1. This last case is especially compelling, as the spacing is 8.5D, which is similar to the space between C1 and D2, and there is a comparable amount of data. However, unlike D1 and D2 when behind the controlled turbine C1, there is no improvement in power production.

For the pairing of turbine C1 and D2 in Fig. 7, the spacing is now 8.5D and wake steering is expected to become more challenging. Yet, the combination yields an improvement in the main wake direction (50 degrees) for the downstream turbine
and the two as a pair. When stepping away from the main wake direction of 50 degrees in either direction, however, things become ambiguous. To the left, we see the amount of data for the baseline case grows smaller and the spread in results for the downstream turbine D2 grows larger (observing the large gray regions). The power is low despite no wake (although this is a wind direction in which the inflow to turbine D2 runs in between turbine R1 and turbine C1.) Therefore, it is probable that lack of data plays a part, and we would expect little change here. To the right, the trend in power goes negative for the downstream
turbine and the pair. Yet, the spread of results completely overlaps with the region of the optimized controller occupying the upper portions of the baseline range.

Finally, observing the results of the turbine C1-D3 pair in Fig. 8, at a spacing of 14D, little improvement is expected and basically none is observed. It is useful to note that turbine D3 has the most data collected and the results seem best converged. As noted earlier, when turbine D3 is waked by turbine R1 at 54 degrees, it is the deeper wake, being 8.5D spacing and no
noticeable change in wake loss occurs, pointing to wake steering being the primary cause of change.



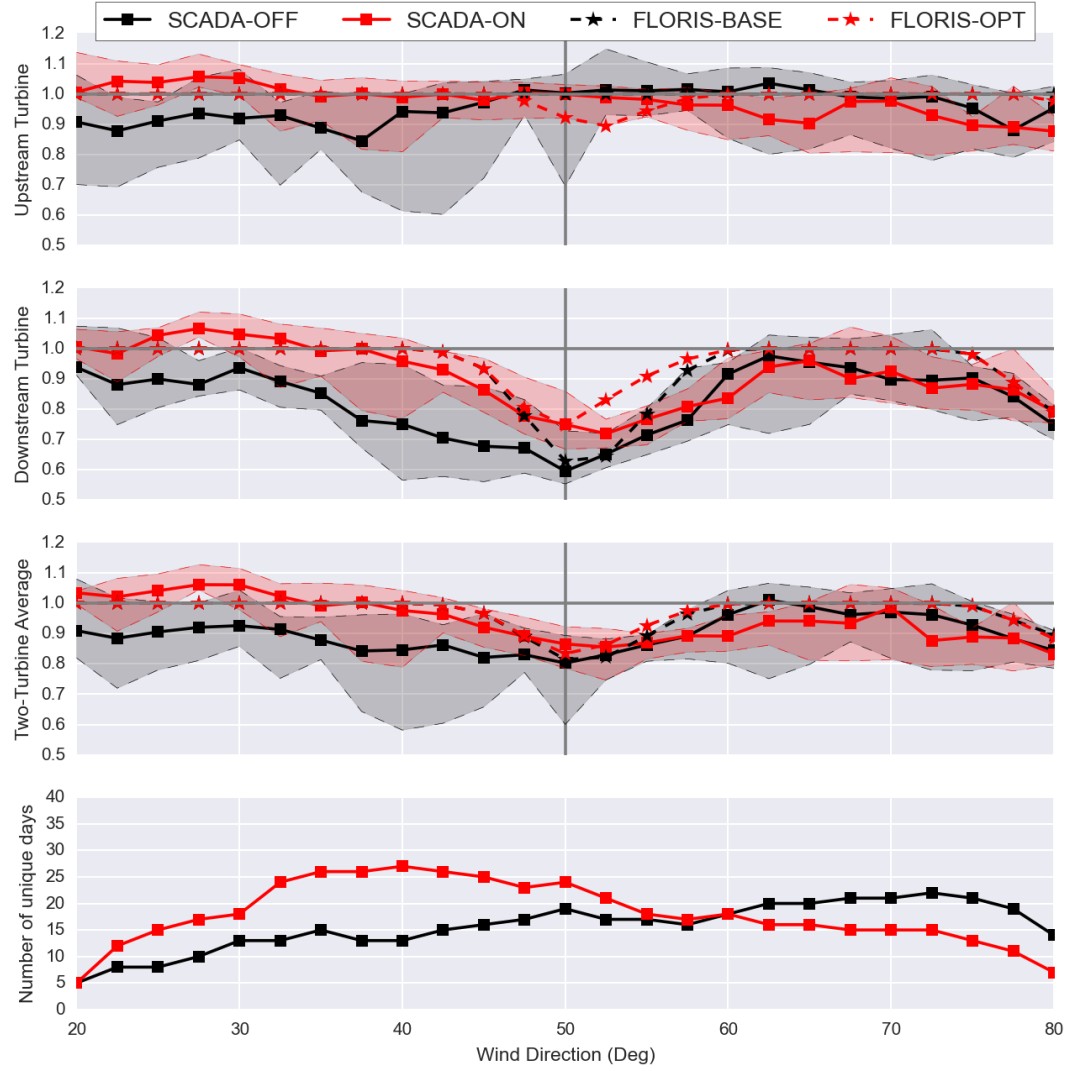

**Figure 7.** This figure shows the results of the turbine C1 to turbine D2 pair. In particular, the top plot shows turbine C1 and the resulting power with the controller on (red) and off (black). The second plot shows the downstream turbine, i.e., turbine D2, with the controller on (red) and off (black). The third plot shows the overall power between the pair of turbines. There is a noticeable power gain with the controller on than off. Finally, the bottom plot indicates the amount of days used to compute the statistics provided in the top three plots.

Also, it is interesting to consider what is happening west of 80 for the downstream turbine D3. FLORIS predicts a return to full power, followed by a dip around 95 degrees when turbine O2 is upstream. However, what is actually observed is a reduction of power basically across the whole range. Considering Fig. 2, this is a range of wind direction without an obvious single turbine wake, but with four turbines still upstream. Notice that unlike an explanation of shallower but more spread loss from wind direction uncertainty, this deficit is deeper than what is predicted by FLORIS. This deficit speaks to an unmodeled





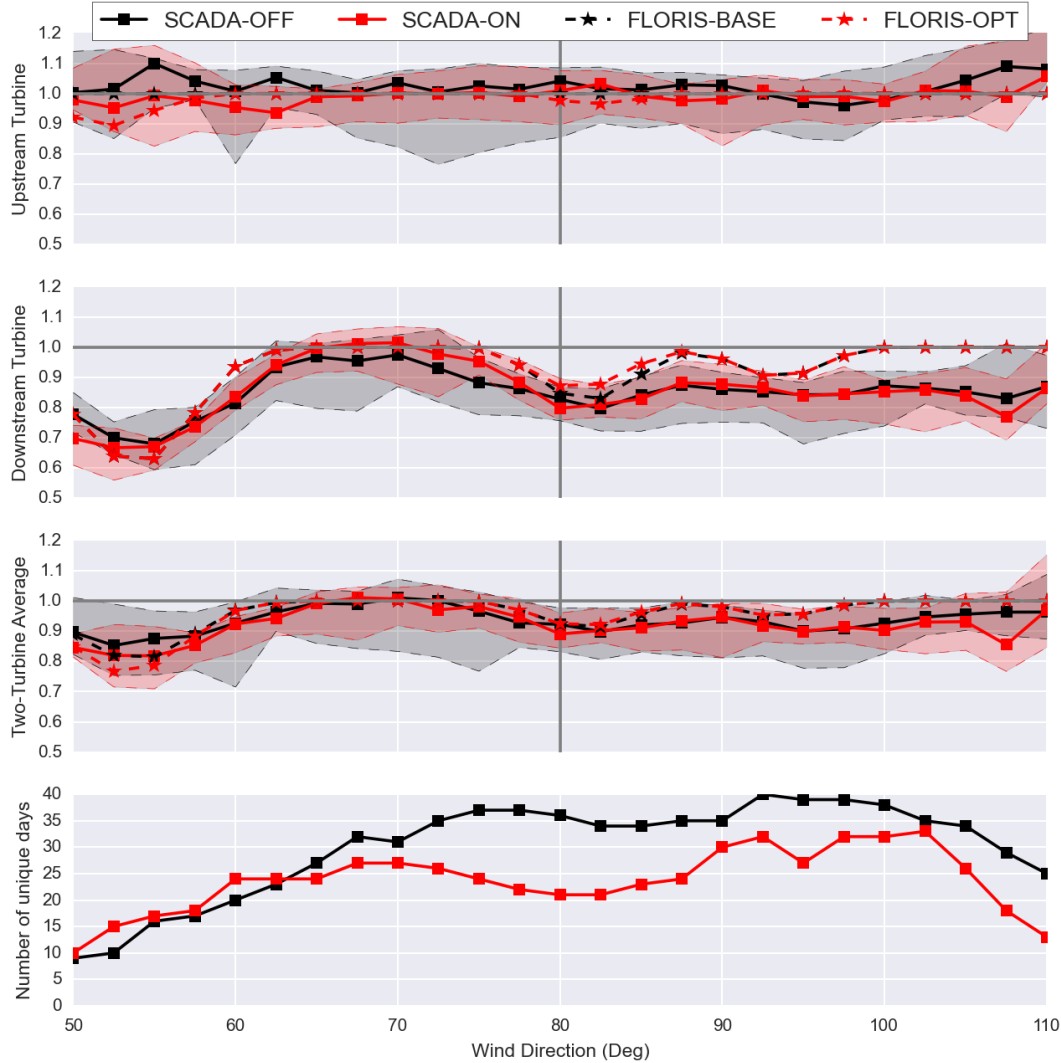

**Figure 8.** This figure shows the results of the turbine C1 to turbine D3 pair. In particular, the top plot shows turbine C1 and the resulting power with the controller on (red) and off (black). The second plot shows the downstream turbine, i.e., turbine D3, with the controller on (red) and off (black). The third plot shows the overall power between the pair of turbines. There is no noticeable power gain with the controller on or off. This is likely because of the increase in spacing between turbines C1 and D3 (14.3D) in comparison to the spacing between turbine C1 and D1 (7D). Finally, the bottom plot indicates the amount of days used to compute the statistics provided in the top three plots.

deep array effect that may prove important to include when FLORIS is used to model and design controllers for multi-turbine arrays.



## 6   Conclusions and Future Work

This study provided several encouraging, albeit qualified results. The main result was that for the directions and spacings (7D and 8.5D) expected to produce an improvement in power for the pair of turbines, such an improvement is observed. The most comprehensible results come from the closer 7D spacing, whereas the 8.5D spacing has some changes that are partly caused by limited data availability. Another good result was the observed agreement between the lower-than-expected power loss with yaw function predicted by SOWFA and the loss derived experimentally. This result is positive, because it provides another form of validation for SOWFA (data sets of utility turbines operating misaligned are not commonly available for testing), and the low power loss value makes wake steering in general more successful.

In the paper, it was discussed that the primary limitations were constraints placed on the amount of data that could be collected at the commercial wind farm, and constraints placed on the wind turbine yaw controller's ability to control. In the case of the pairing with turbine D3, the control and optimal cases are very close in midpoint and range, suggesting that 30-40 separate days of testing is a good target per controller. Had it been possible to toggle between control set points, it is very likely that this number could be reduced. Although, toggling creates some issues of transition. On the controller side, it would appear that the implemented controller was sufficient to secure power gains, however, it could be that a more advanced controller could achieve more.

More generally, the design of closed-loop control systems for wake steering remains an open topic of research. The present method can be regarded as open loop because the actual wake to be controlled is never observed or estimated by the controller. Research that uses lidar to track and control wakes (Raach et al. (2016)) or uses estimation techniques Doekemeijer (2016) may very well improve upon these first results. This is also the subject of ongoing multiyear research projects in the United States (A2e) and Europe (CL-Windcon).

*Acknowledgements.* The authors would like to acknowledge and thank Matthew Churchfield for his support in the use of SOWFA and the development of the simulations. Additionally, the authors thank Pieter Gebraad for his work on this effort during his tenure at NREL.

The U.S. Government retains and the publisher, by accepting the article for publication, acknowledges that the U.S. Government retains a nonexclusive, paid-up, irrevocable, worldwide license to publish or reproduce the published form of this work, or allow others to do so, for U.S. Government purposes.



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
