# Peer review of "Field Test of Wake Steering at an Offshore Wind Farm"

_Wind Energy Science, 2017_

## Referee Comment (RC1) · Anonymous Referee #1 · 10 Feb 2017

Excellent new work on a topical subject.

Page 3 line 3, bad wording: "The positive results motivate further encourage into the design"

Page 5 line 23, losses should be loses

Table 1: "initW D" units should be given, presumably degrees.

Section 4 para 1: "minimal sensitivity to wind speed" - is this likely to depend on Ct, especially above rated? "not much benefit at very high and very low wind speeds" - how can the reader see this - is there a reference?

Section 4 para 2: "for turbine loading and safety reasons, the maximum yaw misalignment was limited to 25 degrees". How was this limit determined? "counter-clockwise

[Figure]

of the wind" - counter-clockwise when viewed from above ? "because it has been demonstrated to be more effective" needs an explanation or reference

Fig. 4: what does the width of the blue band represent?

Section 5, para 2: "reduced to 1-minute averages" - What averaging is used by the yaw controller, and what hysteresis? Does the reference turbine spend significant numbers of 1-minute periods misaligned by a number of degrees until the yaw control kicks in to correct it? Could this affect the conclusions?

Equation 2: Is a cube law actually a good fit, given that there are variable losses etc.? Is N supposed to be a constant, or is it wind speed dependent?

Page 10 line 17: in last sentence of paragraph, it is not very clear what was actually done in FLORIS.

Fig. 6: presumably the plots show the fitted values of 'a'? It seems that SCADA-ON produces significantly more power at the upstream turbine over most of the range. Why, and could this be favourably biasing the result?

Figures 6-8: "the amount of days" should be "the number of days"

General comments:

Turbine loading is barely mentioned, and yet it could be crucial. What are the loading implications on the upstream turbine of the large yaw offsets? How do the downstream turbine loads change? Might they even increase due to partial wake immersion? Maybe the experiment did not include load measurements, but in view of its importance this should at least be mentioned, and any appropriate references provided.

What range of ambient turbulence intensities were experienced? Something should be said about this. It potentially makes a big difference to downstream wake dissipation. It also drives wake meandering, which again is not mentioned but could have significant effects.

---

## Referee Comment (RC2) · Anonymous Referee #2 · 10 Mar 2017

General comments:

This paper presents experimental power production results when considering two-turbine combinations and yawing of the upstream turbine in a commercial offshore wind farm. The paper is reasonably written and publishable as a "discussion paper". I'm rather new to this journal and format, and I'm not sure if one of the goals is to have papers published quickly. The experimental data gathered in this paper are indeed relatively recent and are certainly of interest. However, further analysis perhaps including other data that might be associated with the data already presented would make the paper much stronger. The current paper only presents power data and the trends are not particularly strong. The main results are given in Figures 6 through 8 of the paper. In Figure 6, at the main wind direction of interest (-20 degrees), the yawed (SCADA-OPT) upstream turbine actually produces more power than the non-yawed

(SCADA-BASE) upstream turbine. In Figure 7, these two cases show about the same power production at the main wind direction of interest (50 degrees). A more complete analysis would also evaluate the differences in wind turbine structural loads between the various cases to more fairly assess any associated costs of using wake steering to increase power capture.

Specific comments:

1. Since there are not that many equations and variables in this paper, it would be better to choose more succinct one-letter variables rather than long variable names like "yawLoss" and "initWD". When written in 'math mode', these could represent the product of many variables represented by the individual letters. Even "pP" looks like p times P.

2. In the figures, the power look to be plotted in normalized form. The authors should state what they are normalized to, their own maximum power (as seems to be the case in the "Turbine C1 Power" plot in Figure 3) or some other power reference value (as seems to be the case in most other power plots in the paper).

3. In the figures, units should be given when needed. For instance, the "Turbine C1 Offset" plot in Figure 3 should indicate "(degrees)". And similarly for many other plots in the paper.

4. In general in the figures, use larger font sizes for the axis labels.

5. In the upper plot in Figure 6, it generally looks that "SCADA-BASE" yields more power than "SCADA-OPT". Is there an explanation for this? If you only look at subsets of the data where "SCADA-BASE" and "SCADA-OPT" yield much closer power levels to each other in the upper plot, then are there still the "promising" results for the corresponding subsets for the lower plots?

6. There are several sentences that are quite confusing, and the authors should carefully proofread and make sure that each sentence is easy to understand. For instance,

the sentence on Page 12, lines 30-31: "Yet, the spread of results completely overlaps with the region of the optimized controller occupying the upper portions of the baseline range." After re-reading this several times and looking at Figure 7, I'm still not completely sure what the authors mean. Do they just want to say "Yet, the spread of results of the optimized controller occupies the upper portions of the baseline range." ?

Similarly, the last sentence on Page 12: "As noted earlier, when turbine D3 is waked by turbine R1 at 54 degrees, it is the deeper wake, being 8.5D spacing and no noticeable change in wake loss occurs, pointing to wake steering being the primary cause of change." What does each of the words "change" refer to? Between what and what?

7. How much difference does 1 degree make in whether a wake impacts the downstream turbine? The diagram in Figure 2 indicates that D2 is at 51 degrees relative to C1 and that D3 is 81 degrees relative to C1. Yet, Figures 7 and 8 and the corresponding discussion in the text refer to "50 degrees" and "80 degrees" as the main wind direction to worry about. Further, in discussing Figure 7 looking at the C1-D2 pair, the text on Page 12 even says "To the left" of 50 degrees, "the baseline case grows smaller ... The power is low despite no wake ... " So one degree off, and there is no wake from C1 hitting D2 anymore? How much to the left are the authors really referring to? It might be useful to provide information on how many degrees is needed in each pairing before there is effectively no wake.

Technical corrections:

a. In the last sentence of the introduction, use a different word than "feedback" ... perhaps "evaluation"?

b. Be consistent with variable names. The variables k_d and k_e sometimes appear as K_d and K_e.

c. In Table 1, for parallel structure, perhaps label the Envision turbine as "Envision 4 MW". Also, the values for the variable "initWD" are presumably given in degrees? And

all the other variables are dimensionless?

d. In Figure 3, given the legend, the curve in the lower left plot should be red.

e. In Figures 6 through 8, the legend label "SCADA-OFF" should be "SCADA-BASE" and "SCADA-ON" should be "SCADA-OPT".

f. Figures 6 to 8, lowest plots: I would suggest removing the word "unique" and just use "Number of days" as the y-axis label. I'm not sure what "unique" is meant to indicate. It made me think that if a day was counted for "SCADA-BASE", then it could not be also counted for "SCADA-OPT", though I don't think that is true.

g. Figures 6 to 8, caption: "amount of days" should be "number of days"

h. The ordering of references might be improved. For instance, why is Fleming 2014b not right after Fleming 2014a?

i. Is the Trujillo et al. reference a journal paper, a conference paper, a report, or a personal correspondence?

---

## Author Comment (AC1) · 20 Mar 2017

article [utf8]inputenc [dvipsnames]color

[Figure]

**Response to review 1**

Paul Fleming, Jennifer Annoni

March 20, 2017

We thank the reviewer for their time and suggestions. We have endeavored to respond to all suggestions, which we document here, and accompanying latexdiff (see supplemental pdf) document showing changes (note that the revised figures are included, but not highlighted by latediff).

*Excellent new work on a topical subject.*
Thank you for this comment.
*Page 3 line 3, bad wording: "The positive results motivate further encourage into the design"*
This has been corrected
*Page 5 line 23, losses should be loses*
Corrected
*Table 1: "initW D" units should be given, presumably degrees.*
Fixed

*Section 4 para 1: "minimal sensitivity to wind speed" - is this likely to depend on Ct, especially above rated?*
In the original look-up table produced, there was a wind speed dependency, however,

we noted that the selected yaw offset values were not very different, such as to justify the added complexity of including a wind speed measurement, especially because the measurement of nacelle-mounted anemometer in misaligned conditions might be unreliable.

*"not much benefit at very high and very low wind speeds" - how can the reader see this - is there a reference?*
Figure 5., for example in

Gebraad, P., Thomas, J. J., Ning, A., Fleming, P., Dykes, K. (2016). Maximization of the annual energy production of wind power plants by optimization of layout and yaw-based wake control. Wind Energy. http://doi.org/10.1002/we

Speaks somewhat to this point, the central idea being that at very low wind speeds there is in general very little power, and at very high wind speeds, a waked turbine may still be in above-rated winds, meaning that there is no opportunity for improvement

*Section 4 para 2: "for turbine loading and safety reasons, the maximum yaw misalignment was limited to 25 degrees". How was this limit determined?*
This limit was determined by test engineers at Envision to be at least safe over the span of the experiment. A full set of load cases were run for the conditions expected to occur during the period of the experimentation, and the limit decided such that a good test would be provided for the approach while ensuring no turbine failure consequences to the customer with statistical certainty.

*"counter-clockwise of the wind" - counter-clockwise when viewed from above ?*
That is right, made this clear in text

*"because it has been demonstrated to be more effective" needs an explanation or reference*
Added text and reference to address this point

*Fig. 4: what does the width of the blue band represent?*
Explanation added to legend

*Section 5, para 2: "reduced to 1-minute averages" - What averaging is used by the yaw controller, and what hysteresis? Does the reference turbine spend significant numbers of 1-minute periods misaligned by a number of degrees until the yaw control kicks in to correct it? Could this affect the conclusions?*
This averaging is only done in post-processing. The exact implementation of the yaw controller was not shared with researchers at NREL, however, as mentioned in the paper, it was important for safety reasons that the controller was disabled at sustained winds above 10 m/s with some interpolation between.

*Equation 2: Is a cube law actually a good fit, given that there are variable losses etc.? Is N supposed to be a constant, or is it wind speed dependent?*
A cube law describes the theoretical relation between wind speed and power in below-rated operation, and we include the minimum function to then include the saturation region. N is constant across all turbines, we assume all turbines have the same optimal $C_p$, rotor area and efficiency.

*Page 10 line 17: in last sentence of paragraph, it is not very clear what was actually done in FLORIS.*
Added clarifying text

*Fig. 6: presumably the plots show the fitted values of 'a'? It seems that SCADA-ON produces significantly more power at the upstream turbine over most of the range. Why, and could this be favourably biasing the result?*

This is likely due to several factors. The first is as discussed in earlier section, the loss of power because of yawing is lower on these turbines, and so we would not expect much difference, in other words, noise may be high relative to signal. Further, as is shown in Fig 5, that typical amount of realized offset is less than idealized, further reducing the difference.

For example, the mean offset achieved, according to Fig.5 is around 5 degrees, with the 75% interval ending at 15 degrees, with an $pP$ exponent of 1.41, these would respectively yield power losses of only 99.5% and 95.2%.

If we focus on the region where FLORIS predicts a net increase, from -20 to -12 it is not the case that the upstream turbine power in yaw misalignment always exceeds the power of the baseline case. That said, near -27° for example, there is a clear benefit from this improvement that is not expected as the reviewer points out, and more data would almost definitely revert the trend back to little change.

Finally, given the spread in data, it is also helpful to focus on the banded regions, rather than the specific mid-points, and note that for the most part of the region of highest interest (-20 to -10 where the control is meant to be activated) the bands overlap for the upstream turbine, completely separate for the downstream turbine, and have large non-overlapping regions on net.

*Figures 6-8: "the amount of days" should be "the number of days"*
Fixed

*General comments: Turbine loading is barely mentioned, and yet it could be crucial. What are the loading implications on the upstream turbine of the large yaw offsets?*

*How do the downstream turbine loads change? Might they even increase due to partial wake immersion? Maybe the experiment did not include load measurements, but in view of its importance this should at least be mentioned, and any appropriate references provided.*

Addressing loads was outside the scope of this work, except in ensuring the safe operation of the turbines during the test. However, we are aware of and/or are involved in several related research projects into this question and have added a paragraph of discussion on loads to the conclusion.

*What range of ambient turbulence intensities were experienced? Something should be said about this. It potentially makes a big difference to downstream wake dissipation. It also drives wake meandering, which again is not mentioned but could have significant effects.*

Turbulence intensity is a crucial part in analyzing wake steering strategies. In this study, we did not include turbulence intensity as a parameter for wake steering, because it was not included in the engineering model FLORIS used to design the strategy. However, in the future, this will be necessary to maximize the benefit of the wake steering strategy. A new version of FLORIS under development does account for the effect of turbulence intensity, and even added turbulence intensity, however, this was not ready at the time of the experiment.

It should as be reiterated that this is an offshore wind plant and turbulence intensity is generally lower than onshore wind plants.

---

## Author Comment (AC2) · 20 Mar 2017

article [utf8]inputenc [dvipsnames]color

[Figure]

**Response to review 2**

Paul Fleming, Jennifer Annoni

March 20, 2017

We thank the reviewer for their time and suggestions. We have endeavored to respond to all suggestions, which we document here, and accompanying latexdiff document (see supplemental pdf) showing changes (note that the revised figures are included, but not highlighted by latediff).

*General comments: This paper presents experimental power production results when considering two turbine combinations and yawing of the upstream turbine in a commercial offshore wind farm. The paper is reasonably written and publishable as a "discussion paper". I'm rather new to this journal and format, and I'm not sure if one of the goals is to have papers published quickly. The experimental data gathered in this paper are indeed relatively recent and are certainly of interest. However, further analysis perhaps including other data that might be associated with the data already presented would make the paper much stronger.*

Thank you for your frank feedback. We unfortunately are not able to collect further data as this project has ended. We also recognize that because the experiment was run at a commercial offshore wind farm with various constraints, all the sensing we could have wanted was not available, and the turbines could not be controlled in the fashion (with control alternating regularly on and off) and could not be accommodated at this time.

We have attempted to make clear acknowledgments of the limitations of these results in the text.

We do however hope to show value in providing these results from a commercial wind farm.

*The current paper only presents power data and the trends are not particularly strong. The main results are given in Figures 6 through 8 of the paper. In Figure 6, at the main wind direction of interest (-20 degrees), the yawed (SCADA-OPT) upstream turbine actually produces more power than the non-yawed (SCADA-BASE) upstream turbine.* This is likely due to several factors. The first is as discussed in earlier section, the loss of power because of yawing is lower on these turbines, and so we would not expect much difference, in other words, noise may be high relative to signal. Further, as is shown in Fig 5, that typical amount of realized offset is less than idealized, further reducing the difference.

For example, the mean offset achieved, according to Fig.5 is around 5 degrees, with the 75% interval ending at 15 degrees, with an $pP$ exponent of 1.41, these would respectively yield power losses of only 99.5% and 95.2%.

If we focus on the region where FLORIS predicts a net increase, from -20 to -12 it is not the case that the upstream turbine power in yaw misalignment always exceeds the power of the baseline case. That said, near -27° for example, there is a clear benefit from this improvement that is not expected as the reviewer points out, and more data would almost definitely revert the trend back to little change.

Finally, given the spread in data, it is also helpful to focus on the banded regions, rather than the specific mid-points, and note that for the most part of the region of highest interest (-20 to -10 where the control is meant to be activated) the bands overlap for the upstream turbine, completely separate for the downstream turbine, and have large non-overlapping regions on net.

*In Figure 7, these two cases show about the same power production at the main wind direction of interest (50 degrees).*

The caption of this plot indicated power was not correct and has been improved. A turbine spacing of 8.5 diameters is at a range where we expect wake-steering to be a challenge, even in idealized simulations. The text describing the figure gives a more nuanced description.

*A more complete analysis would also evaluate the differences in wind turbine structural loads between the various cases to more fairly assess any associated costs of using wake steering to increase power capture.*

Addressing loads was outside the scope of this work, except in ensuring the safe operation of the turbines during the test. However, we are aware of and/or are involved in several related research projects into this question and have added a paragraph of discussion on loads to the conclusion.

*Specific comments:*
*1. Since there are not that many equations and variables in this paper, it would be better to choose more succinct one-letter variables rather than long variable names like "yawLoss" and "initWD". When written in 'math mode', these could represent the product of many variables represented by the individual letters. Even "pP" looks like p times P.*

In this case, if it's ok with the reviewer, we prefer the more expressive, long, notation, since as you say, there is not that many equations. $pP$ is used for consistency with other papers.

*2. In the figures, the power look to be plotted in normalized form. The authors should*

*state what they are normalized to, their own maximum power (as seems to be the case in the "Turbine C1 Power" plot in Figure 3) or some other power reference value (as seems to be the case in most other power plots in the paper).*
For Figure 3, explanatory text has been added to the caption.

For figures 6-8, text has been added to the explanation of equation 2 to more clearly describe the fitting and normalization process.

*3. In the figures, units should be given when needed. For instance, the "Turbine C1 Offset" plot in Figure 3 should indicate "(degrees)". And similarly for many other plots in the paper.*
Fixed.

*4. In general in the figures, use larger font sizes for the axis labels.*
Fixed.

*5. In the upper plot in Figure 6, it generally looks that "SCADA-BASE" yields more power than "SCADA-OPT". Is there an explanation for this? If you only look at subsets of the data where "SCADA-BASE" and "SCADA-OPT" yield much closer power levels to each other in the upper plot, then are there still the "promising" results for the corresponding subsets for the lower plots?*
For similar reasons as the responses above on this topic, In figure 6., in the region from -20 to -12 where the controller is mostly expected to be active, the upstream turbine bands basically overlap, which could be interpreted as not significant change given data, where the downstream bands are completely un-overlapped – a significant change –, yielding a partial overlap between the bands on the total power.

In the same figure, around -27 degrees, indeed since the controller is not expected to be operating, the improvement is most likely due to the variation occuring in the

upstream turbine..

*6. There are several sentences that are quite confusing, and the authors should carefully proofread and make sure that each sentence is easy to understand. For instance, the sentence on Page 12, lines 30-31: "Yet, the spread of results completely overlaps with the region of the optimized controller occupying the upper portions of the baseline range." After re-reading this several times and looking at Figure 7, I'm still not completely sure what the authors mean. Do they just want to say "Yet, the spread of results of the optimized controller occupies the upper portions of the baseline range." ?*

This sentence in particular has been re-written, and more generally, the paper will pass through an additional independent review to check style more deeply.

*Similarly, the last sentence on Page 12: "As noted earlier, when turbine D3 is waked by turbine R1 at 54 degrees, it is the deeper wake, being 8.5D spacing and no notice-able change in wake loss occurs, pointing to wake steering being the primary cause of change." What does each of the words "change" refer to? Between what and what?*

Thank you for pointing this out, with this I had hoped to make the point that similarly to C1-D2, this is a wake loss with a distance of 8.5D, except that in this case there is no wake steering applied. In this case the power of the waked turbine is not changed between the baseline and optimized case, which suggests that the change in downstream turbine power in earlier cases, can be attributed largely to wake steering and less to seasonal variation.

The text has been revised.

*7. How much difference does 1 degree make in whether a wake impacts the downstream turbine? The diagram in Figure 2 indicates that D2 is at 51 degrees relative to*

*C1 and that D3 is 81 degrees relative to C1. Yet, Figures 7 and 8 and the correspond-*
*ing discussion in the text refer to "50 degrees" and "80 degrees" as the main wind*
*direction to worry about. Further, in discussing Figure 7 looking at the C1-D2 pair, the*
*text on Page 12 even says "To the left" of 50 degrees, "the baseline case grows smaller*
*... The power is low despite no wake ... " So one degree off, and there is no wake from*
*C1 hitting D2 anymore? How much to the left are the authors really referring to? It*
*might be useful to provide information on how many degrees is needed in each pairing*
*before there is effectively no wake.*

The data has been binned to 5-degrees, and so 1-degree increments are often within-
bin, and within the probably margin of error of a nacelle wind vane.

The statement to the left means starting from 50 degrees and progressing leftward.

*Technical corrections: a. In the last sentence of the introduction, use a different word*
*than "feedback" ... perhaps "evaluation"?*

Fixed

*b. Be consistent with variable names. The variables $k_d$ and $k_e$ sometimes appear as*
*$K_d$ and $K_e$.*

All set to $k_e$ and $k_d$.

*c. In Table 1, for parallel structure, perhaps label the Envision turbine as "Envision 4*
*MW". Also, the values for the variable "initWD" are presumably given in degrees? And*
*all the other variables are dimensionless?*

Changes all made, non-labeled are dimensionless

*d. In Figure 3, given the legend, the curve in the lower left plot should be red.*

That is true, this is fixed.

*e. In Figures 6 through 8, the legend label "SCADA-OFF" should be "SCADA-BASE" and "SCADA-ON" should be "SCADA-OPT".*
This is fixed.

*f. Figures 6 to 8, lowest plots: I would suggest removing the word "unique" and just use "Number of days" as the y-axis label. I'm not sure what "unique" is meant to indicate. It made me think that if a day was counted for "SCADA-BASE", then it could not be also counted for "SCADA-OPT", though I don't think that is true.*
The point of the word unique, is to stress that these are not number of days, as in, 24-hour periods, but since the fits were made on a per-day basis, how many separate days contain the necessary condition to construct such a fit? If this were changed to "number of days", the fear is misinterpretation of for example 10 days implies 240 hours, which it does not.

*g. Figures 6 to 8, caption: "amount of days" should be "number of days"*
Fixed

*h. The ordering of references might be improved. For instance, why is Fleming 2014b not right after Fleming 2014a?*
The provided journal latex class determines the order, I'm not sure how it is done.

*i. Is the Trujillo et al. reference a journal paper, a conference paper, a report, or a personal correspondence?*
Journal article, this citation has been improved

**Supplement:**

[revised manuscript text omitted]

---

## Author Response (AR1)

April 4, 2017

Dear Editors,

  Please find attached a revised version of the paper.  The paper includes the revisions discussed in the interactive discussions, and matches the latex-diff changes indicated in the supplementary files of those discussions.

Paul Fleming

Paul Fleming

[revised manuscript text omitted]

---

## Author Response (AR2)

April 12, 2017

Dear Editors,

Thank you for your comments.  We have endeavored to make all changes requested by the chief editor.

The only change we ask permission to reconsider is within the third point.  We made the recommended changes to the notation for yaw loss and initial wind direction, however, "pP" we have now used in several past publications, for example, (Gebraad, el al. (2014). "Wind plant power optimization through yaw control using a parametric model for wake effects-a CFD simulation study." Wind Energy).

For consistency with previous publications, and the notation used within the FLORIS software, we ask that this notation be allowed.  Thank you for your consideration.

Paul Fleming

Paul Fleming
Senior Engineer | National Wind Technology Center

National Renewable Energy Laboratory (NREL)
15013 Denver West Parkway | Golden, CO 80401
303-384-6918
paul.fleming@nrel.gov | www.nrel.gov